# Reasoning Models are Test Exploiters: Rethinking Multiple Choice

**Narun K. Raman** [1]   **Taylor Lundy** [1]   **Kevin Leyton-Brown** [1]

## Abstract

When evaluating Large Language Models (LLMs) in question-answering domains, multiple-choice question answering (MCQA) is widely used because it enables automatic grading. However, MCQA also exposes models to answer options that can be exploited in ways that inflate reasoning ability. We study this phenomenon across 15 question-answering benchmarks and 27 LLMs by systematically varying how and when models are exposed to answer options. For non-reasoning LLMs, MCQA can remain a good proxy for free-text performance when any chain-of-thought is produced only before the options are revealed. However, this "decoupled" format is not realizable for most reasoning models: they are designed to emit reasoning tokens whenever they are prompted, so if options are present they inevitably "reason over" the options. In practice, this makes reasoning models particularly effective at extracting signal from options, and can create large, misleading gains over free-text baselines. To characterize how models exploit MCQA, we introduce diagnostic probes that isolate option-only and question-plus-option exploitation pathways, and we quantify how design choices such as distractor strength and "none-of-the-above" answers effect exploitability. Finally, we examined the practice of multiple choice as an error diagnostic: inferring a model's mistake from the wrong option it picks. On benchmarks where reasoning can be expressed as code, we ask models to output code, we then executed it varying the inputs, and compared the resulting input–output behavior, revealing failure modes that MCQA diagnostics obscure. Lastly, we offer practical guidelines when analyzing results from MCQA that better reflect LLMs' genuine reasoning capabilities.

[1]Department of Computer Science, University of British Columbia, Vancouver, BC. Correspondence to: Narun K. Raman <narunram@cs.ubc.ca>.

*Proceedings of the 43rd International Conference on Machine Learning*, Seoul, South Korea. PMLR 306, 2026. Copyright 2026 by the author(s).

## 1. Introduction

Early work in machine comprehension adopted multiple-choice question answering (MCQA) for straightforward, automatic grading and to mirror familiar exam formats. The MCTest corpus introduced this paradigm with 660 children's stories and four-option questions, demonstrating that constraining answers to a fixed label set avoids free-text ambiguity and simplifies evaluation (Richardson et al., 2013). Successors such as RACE and ARC expanded scale and domain coverage (Lai et al., 2017; Clark et al., 2018), while MMLU broadened to 57 subjects for measuring general knowledge and reasoning in a multiple-choice format (Hendrycks et al., 2021a). MCQA benchmarks have now been widely adopted for LLM evaluation (Liang et al., 2022; Li et al., 2024), with benchmarks like MMLU (Hendrycks et al., 2021a), GPQA (Rein et al., 2023), and ARC (Clark et al., 2018) having emerged as standard performance yardsticks. While high accuracy on MCQA benchmarks have historically been a good signal of reasoning (e.g., GPT-4 achieves $88.7\%$ accuracy on MMLU, outperforming smaller models like Gemma (7B) which achieved $66.0\%$), the strength of that signal has been recently called into question. Performance gains have various causes: in part, LLMs truly improve at downstream tasks, and in part, they benefit from training on the benchmarks that are used to evaluate them.

A third reason for performance gains has been getting increasing attention: the MCQA format can give models an opportunity to exploit the structure of the test itself. Models can exploit elimination heuristics or statistical "artifacts" in the option text, even when the question is withheld, achieving well above chance on purely answer-only inputs (Balepur et al., 2024; Myrzakhan et al., 2024). Permuting or randomizing option positions reveals selection biases that debiasing methods (e.g., PriDe) must address (Zheng et al., 2024). Turner & Kurzeja (2025) find that a decision tree can reach almost $80\%$ on TruthfulQA without even reading the question. Complementing this, there is recent work demonstrating that introducing a "None-of-the-Above" option can disrupt performance for LLMs (Raman et al., 2024; 2025; Tam et al., 2025), linking option design explicitly to inflated scores. However, robustness varies widely: when distractors are strengthened or randomized, certain instruction-tuned models maintain unexpectedly stable performance, as explored by Wang et al. (2024a). Most relevantly, recent work

by Raman et al. (2025) observed models boosting MCQA performance via "plug-and-chug" tactics on provided options. This limitation is most consequential when MCQA is used as a proxy for open-ended reasoning or generation: when the downstream task itself requires selecting among explicit alternatives (e.g., classification), option-aware reasoning may be appropriate.

Beyond straightforward grading, MCQA is also often treated as a diagnostic tool: when a model answers incorrectly, the selected distractor is taken as evidence about the model's underlying misconception (Haladyna et al., 2002; Liu et al., 2025). However, such a diagnostic reading is suspect. With only a few candidates to choose from, different reasoning processes can end up selecting the same option, and the same underlying incorrect reasoning process can correspond to different distractors when the options differ in small, surface-level ways. Empirically, this sensitivity is well documented: reordering the options or making tiny edits can change which distractor a model selects even when its underlying reasoning has not changed, suggesting that distractor identity often reflects surface-level choice behavior rather than a stable internal misconception (Pezeshkpour & Hruschka, 2024; Zheng et al., 2024).

Despite these issues and others,[1] many leaderboards and model releases continue to emphasize MCQA tasks. For instance, MCQA comprises 3 out of 4 datasets in o1-preview's blog post on "Learning to Reason with LLMs" (OpenAI, 2024), in 66 % of tasks in Meta's announcement of Llama 3.1 (Meta, 2024), and 32 % of tasks in HELM (Perlitz et al., 2024). Conversely, studies of real-world usage indicate a stark contrast: queries from ShareGPT's dataset show users predominantly asking free-form generative outputs rather than validation tasks; MCQA-style queries constitute merely 7.2 % of the tasks (Ouyang et al., 2023).

A popular recent approach to "fixing" MCQA expands the option set with tougher distractors (Wang et al., 2024c; Gema et al., 2024). Other benchmarks go further towards true free-text question answering (FTQA), designing entirely free-response benchmarks (Myrzakhan et al., 2024). Span-extraction benchmarks such as SQuAD (Rajpurkar et al., 2016), HotpotQA (Yang et al., 2018), and DROP (Dua et al., 2019) require models to locate answer spans in passages and are evaluated by exact-match or token-level F1. In mathematics, GSM8K (Cobbe et al., 2021) and MATH (Hendrycks et al., 2021b) use free-form numeric or short-text answers with normalization-based exact match; in program synthesis, HumanEval (Chen et al., 2021) and MBPP (Austin et al., 2021) rely on execution-based unit tests as judges. Nonetheless, FTQA remains feasible primarily in constrained domains where responses are numeric or

easily parsed (e.g., Hendrycks et al., 2021b; Raman et al., 2025). Motivated by this tension, researchers have proposed promising LLM-based evaluation protocols that aim to move beyond standard MCQA (Kočiský et al., 2018; Li et al., 2023; Chandak et al., 2025).

Rather than proposing another fix, we treat MCQA as a measurement instrument and ask what it is actually measuring. We formalize *exploitation* as the excess accuracy attributable to seeing answer options—i.e., performance gains that arise in the multiple-choice format beyond what the same model achieves when answering the question without options. We then localize when this excess arises by isolating the effects of CoT timing and option design.

We further argue that MCQA is a weak diagnostic instrument: because it collapses behavior to a single option choice, it cannot distinguish errors in a model's underlying computation from artifacts of the final selection step. As a complementary diagnostic, we also evaluate models in FTQA and prompt them to output executable CoT, allowing us to test whether a model's stated reasoning reproduces its answer and whether distinct models share the same underlying failure mode.

We begin by describing the benchmarks we selected in Section 2 and then go on to describe our evaluation methodology and model lineup in Section 3. We compare model performance across five evaluation formats: (1) MC-CoT, where models must choose among a set of options with no question provided; (2) QMC-CoT, where models are given a question and asked to choose among fixed answer options; (3) Q-CoT, where models produce entirely free-form answers to a question without provided choices; (4) Q-CoT-MC-1T, where models first reason freely before selecting from provided options; and (5) where a placeholder none of the above (NOTA) option[2] is introduced to calibrate the evaluation baseline and reduce reliance on elimination strategies. Finally, to probe diagnostic validity beyond option choice, we conduct a targeted analysis on math and code-reasoning tasks in which models are asked to output executable reasoning traces. By running these traces on counterfactual inputs, we clustered models that share the same erroneous reasoning. In total, we spent $2,146.51 making requests to OpenAI's API and 4.92 GPU years of compute to evaluate open-weight models. We then examine performance differences in accuracy to expose how MCQA can inflate perceived strengths when LLMs are allowed to reason over the options. We leave the discussion of these findings in Section 4, and offer a few highlights. When options precede CoT (QMC-CoT), reasoning models gain substantially over Q-CoT even accounting for post-hoc "closest-answer"

---

[1]There is evidence that MCQA can *deflate* scores (Wang et al., 2024b;a; Molfese et al., 2025), we focus on inflationary effects.

[2]We randomized the order of options so "above" is not a useful indicator. We therefore used "No other option is correct" but for intuition's sake we refer to it in the paper as NOTA.

mapping, revealing a distinct second-pass shortcut; a NOTA intervention dampens this shortcut and narrows the gap between reasoning and non-reasoning models, while making option sets "harder" does not reliably curb exploitability and, for some models, can even increase it. Through behavioral probing, we find evidence of both universal pitfalls and family-specific biases: models within the same family are $2.7 \times$ more likely to share an error pattern than models from different families, yet most models share at least one incorrect behavior with $94.7\%$ of other models. Finally, we offer some practical takeaways when designing benchmarks in Section 5.

## 2. Benchmarks

We evaluated LLMs on 15 benchmarks spanning diverse domains and question formats. Except where indicated otherwise, each benchmark consists entirely of four-option multiple-choice questions.

**MCQA Benchmarks**  We evaluate on MMLU and MMLU-Pro (Hendrycks et al., 2021a; Wang et al., 2024c). We also use the Open-LLM suite spanning standard MCQA datasets (e.g., ARC, WinoGrande, PIQA, CommonsenseQA, RACE, MedMCQA, OpenBookQA) (Myrzakhan et al., 2024), and GPQA Diamond, the hardest split of GPQA with graduate-level science questions (Rein et al., 2023).

**FTQA Benchmarks**  For FTQA we use GSM8K (grade-school math), MATH (competition math), PythonIO (program-output prediction from HumanEval and MBPP), and STEER-ME (economics questions) (Cobbe et al., 2021; Hendrycks et al., 2021b; Chen et al., 2021; Austin et al., 2021; Zhang et al., 2024; Raman et al., 2025).

### 2.1. Question Format Conversion

We aim to examine how format alone affects performance; this section describes how we converted the MCQA benchmarks listed above to FTQA and vice versa.

**MCQA → FTQA:**  We start with the datasets within Open-LLM. The dataset suite was constructed by filtering out questions from multiple datasets, which were not suitable for open-style answering. The filtering process they used kept many MCQA questions that would not be viable FTQA questions. So we employed two subsequent filtering procedures: (1) Removed all questions that contained text that explicitly or implicitly mentioned the options in the stem (e.g., 'Which of the following', 'What can be concluded from the passage') via substring search, and (2) Removed all stems that did not end with a period or question mark (e.g., 'While training the rats, the trainers have to be'). We also removed questions which contained options

with any NOTA-like text (e.g., "none of the following") so that our reported NOTA results come from a controlled intervention that we introduce ourselves. After this filtering process, $62.81\%$ of the total dataset remained of both MCQA/FTQA questions. For more details and a breakdown for each dataset, see Figure 7 in the appendix. Note that this likely omitted convertible MCQA questions. We did the same filtering for MMLU-Pro, reducing the original test set of $12{,}032$ questions to $7{,}130$ questions.

**FTQA → MCQA:**  For most of the datasets (all but STEER-ME) that were originally instantiated in FTQA as listed in Section 2, we used the MCQA versions created by Zhang et al. (2024). These datasets were constructed by collecting answers and incorrect predictions on GSM8K, MATH, HumanEval and MBPP from 60 open-weight models. Finally, STEER-ME includes programmatically generated multiple-choice options as part of the benchmark.

We do a final filtering pass, running our grading function over the correct answers to check whether they can be converted into a grade-able format. We call questions that pass this filtering step CoT-extractable.

## 3. Methodology

Our objective is to measure how much of an LLM's MCQA performance reflects genuine problem solving versus exploitation. We first specify the evaluation formats (inputs and allowed responses), then define one- and two-stage configurations built from them; next we describe our evaluation metrics (accuracy and exploitation), MCQA↔FTQA conversions, and experimental setup.

### 3.1. Evaluation Formats

The question of how to present MCQA and FTQA questions to LLMs gives rise to a large design space. We focus on two key dimensions of this space: how the question is formatted and what form the LLM's response is allowed to take.

**Question formats**  We present MCQA questions to the model in three formats. In the first format, we present only the $k$ multiple-choice options for a given question, hiding the actual question stem (i.e. "What is 2+2?"). This format intends to identify the amount of exploitable information that is present in the options themselves, similar to the work by Balepur et al. (2024); Chandak et al. (2025). In the second format, we present the question stem followed by its $k$ options.

Following work by Raman et al. (2024; 2025); Tam et al. (2025), in the third format, we amended multiple-choice questions by inserting a "None of the above" (NOTA) placeholder. For a given benchmark in $1/k$ of the questions, we

replaced the *correct* answer with NOTA. In the remaining questions, we replaced one *incorrect* answer with NOTA, chosen uniformly at random.

We format every MCQA question in our analysis into these four formats:

| Format | Model input ($s$) |
|---|---|
| MC | Only multiple choice options |
| MCNA | Same as MC but with NOTA as an option |
| QMC | Question and multiple choice options |
| QMCNA | Question with NOTA as an option |

*Table 1.* What the model *receives*.

| Configuration | Description |
|---|---|
| MC-CoT | The model is given just MC and outputs a CoT response. |
| MCNA-CoT | The model is given just MCNA and outputs a CoT response. |
| Q-CoT | The model is given the question and outputs a CoT response. |
| QMC-CoT | The model is given the question w/ MC and outputs a CoT response. |
| QMCNA-CoT | The model is given the question w/ MCNA and outputs a CoT response. |

*Table 2.* One-stage evaluation configurations.

**Response formats** We consider how an LLM responds to some context as a function mapping an input string to an output string or a distribution over next tokens. Exactly what this function outputs not only depends on the inputted context but also on the LLM. Reasoning models (e.g., OpenAI's o-series; DeepSeek's R1) are fine-tuned always to output chain-of-thought tokens; we denote any response format where there is chain-of-thought before an answer as CoT. Non-reasoning models can be prompted to output a single token without *any* chain-of-thought reasoning; we denote such a response format as 1T.

We follow Wang et al. (2024a;b) and explicitly tell the model to output a single token to prevent mismatch between the letter obtained in 1T and CoT. See Appendix A for our exact prompt.

**Evaluation configurations** An evaluation configuration is an (input, response) pair which when called produces an output that can be evaluated. We consider both one-stage and two-stage configurations. We begin by defining one-stage configurations and then use those concepts to define our two-stage configurations.

MCQA and FTQA are standard one-stage evaluation configurations. An important design dimension separating MCQA and FTQA is whether a model can incorporate the options into their reasoning (QMC-CoT) or whether the reasoning happens without knowledge of the options (Q-CoT). We consider five one-stage evaluation configurations:

Note that MC-CoT is similar to the methodology introduced by Balepur et al. (2024), however, while they restrict the LLM to the 1T response function, we are interested in the effect of reasoning over the options and so restrict to the CoT response function.

Given these one-stage configurations we can also construct two-stage configurations which first ask the model to perform a Q-CoT step, after which the model is presented with

the options and is asked to answer with CoT or 1T. Raman et al. (2024) introduced Q-CoT-MC-1T (f.k.a. "hidden"); a two-stage configuration in which the second response is 1T. However, the response function used in the second step depends on the LLM as reasoning models cannot respond with 1T. Table 3 describes the four two-stage configurations we consider.

| Configuration | Stage-2 description after Q-CoT |
|---|---|
| Q-CoT-MC-1T | Given context + MC, output 1T |
| Q-CoT-MCNA-1T | Given context + MCNA, output 1T |
| Q-CoT-MC-CoT | Given context + MC, output CoT |
| Q-CoT-MCNA-CoT | Given context + MCNA, output CoT |

*Table 3.* Two-stage evaluation configurations.

A notable limitation is that because the second stage reintroduces the options to the same model that generated the chain-of-thought, reasoning models can still exploit option artifacts or apply elimination heuristics when selecting their final label. This means that any two-stage configuration serves primarily as a useful measure of exploitation for non-reasoning models. However, Q-CoT-MCNA-CoT still offers insight into the ability of reasoning models to exploit the options. Models only get to reason on $1 - 1/k$ questions where the correct answer is present in the second-step option sets, meaning relying on elimination, rather than grounding their answers in the earlier reasoning trace, is more likely to fail.

### 3.2. Executable CoT Traces for Diagnostic Probing

We performed an additional diagnostic analysis on a subset of *CoT-extractable questions*, those in which a model's intermediate reasoning can be represented as an executable procedure. For these tasks, we prompted models to output Python code that computes the answer, using a strictly enforced output format (Appendix A). Much like asking a human to "show their work." The outputted code is treated as an executable reasoning trace: a candidate function map-

| Evaluation Configuration | Model Type | |
| --- | --- | --- |
| | Reasoning | Non-reasoning |
| MC-CoT | ✓ | ✓ |
| Q-CoT | ✓ | ✓ |
| QMC-CoT | ✓ | ✓ |
| QMCNA-CoT | ✓ | ✓ |
| Q-CoT-MC-1T | ✗ | ✓ |
| Q-CoT-MCNA-1T | ✗ | ✓ |
| Q-CoT-MC-CoT | ✓ | ✗ |
| Q-CoT-MCNA-CoT | ✓ | ✗ |

*Table 4.* This table lists the evaluation protocols we ran for each model type.

ping problem inputs to outputs.

The generated program is executed in a restricted Python sandbox (standard library only) with a 5-second timeout. To probe behavior beyond the original instance, we executed the same program under alternative inputs with new values for the input variables already present in the code (e.g., numeric quantities defined prior to function call). This allowed us to evaluate the other inputs without re-prompting.

**Probe generation and function extraction.** For math tasks, we extract function calls from the generated code using AST parsing, first searching for ANSWER = solve(...) patterns, then falling back to any call to the target function (e.g., solve). Argument values are resolved by evaluating literals and resolving variable references from preceding assignments. For code tasks, probe inputs are derived from unit test templates, extracting function call arguments from test assertions.

Probe values are generated deterministically using a seeded random number generator: for each original argument value, the seed combines the sample ID, the normalized value, the probe index (0 to 199), and a global seed parameter. This ensures that the same argument receives the same probe value across all models, even when models use different argument orderings. See Appendix D for more details on input types.

**Behavior hash computation.** The behavior hash is computed as the SHA-256 hash of the JSON-serialized probe outputs, with floating-point values rounded to 6 decimal places to avoid precision-based differences. Two traces are considered behaviorally equivalent if their hashes match exactly, indicating identical input–output mappings across all 200 probe instances (up to the same normalization used for ground-truth comparison, which handles LaTeX expressions, fractions, and other mathematical representations).

### 3.3. Evaluation Metrics

**Accuracy:** The primary metric is the percentage of questions answered correctly. For configurations with options this is simple: a response is correct if the model's chosen option letter matches the correct option letter. For those without, a response is correct if it matches the known correct answer. In the case of numeric answers, we require numerical equivalence after rounding the correct answer to the number of significant figures the model reports. This penalizes overprecision: if an LLM reports more significant figures than necessary and is incorrect, that discrepancy is treated as an error. Note that even when relaxing this and rounding to the answer significant figures did little to change the results. For functional answers, we convert the text into sympy and simplify, testing equivalence through sympy's built-in functionality. See Appendix C.1 for the exact Python grading function.

**Exploitation:** This measures the excess accuracy a model achieves from having access to answer options. We define exploitation as the additive gap between accuracy with options and accuracy without options. However, raw accuracy with options is inflated even in the absence of any reasoning, since a model can always achieve non-zero accuracy by random guessing. To isolate accuracy attributable specifically to option-dependent behavior we use a chance model for how a model selects among options when it cannot solve the question.

A simple chance model assumes uniform guessing: for each question with $k$ options, let $A_{MC}$ be the model's QMC-CoT accuracy, $A_{FT}$ its Q-CoT accuracy, and $1/k$ the random-guess baseline: $E = A_{MC} - \left( A_{FT} \cdot \left( \frac{k-1}{k} \right) + \frac{1}{k} \right).$ A positive value for $E$ means the LLMs correctness above chance while seeing options exceeds what it can do without options; in other words, accuracy that relies on the options rather than underlying knowledge. The units are percentage points: $E = 0.12$ means 12 extra correct answers per 100 questions that vanish when the options are withheld. We

consider more sophisticated chance models in Section 4.

## 3.4. Experimental Setup

In total, we evaluated 27 LLMs. We leave the full list including the model cards and configurations to Table 5 in Appendix B. Table 4 lists the evaluation configurations that we ran on each model type. We obtained 23 open-weight LLMs from the HuggingFace Hub (Wolf et al., 2019) and ran them on 1 to 4 L40 GPUs. We used OpenAI's API for o3 and gpt-4o. For all prompts, we set the softmax temperature $T$ to recommended settings; greedy decoding ($T = 0$) for non-reasoning models and $T = 0.6$–$0.8$ for reasoning models.

For all of the datasets, other than STEER-ME, we evaluated the open-weight LLMs on 5,000 questions per dataset and the closed-weight LLMs (o3 and gpt-4o) on 1,000 questions per dataset. For STEER-ME, we ran all open-weight LLMs on 100 questions per element and the closed-weight models on 20 questions per element, resulting in 5,800 and 1,160 questions in total, respectively.

**Answer Extraction** For CoT we ask the model to leave the answer in '\boxed{}.' To extract answers from the model-generated reasoning content we use regex and match until we find the correct closing brace. If this regex fails to retrieve a valid response, we use a secondary regex '\.*\[aA\]nswer:\s*\([^{}]+\)' for a second attempt to extract the answer. For 1T we decode the distribution over the next token after 'Answer: ' as well as 'Answer:\n', picking whichever assigns the correct token the highest probability.

## 4. Results

Figure 1 reports each LLM's pass@1 accuracy under the QMC-CoT format and the Q-CoT formats. A clear trend emerges: The largest models—and the most performant— exhibit the largest positive gaps between QMC-CoT and Q-CoT (see Figure 2). All models above roughly 50 B parameters scored 30 to 40 percentage points higher when choices are given before CoT, with the difference being even larger for reasoning models. One might expect that a sufficient rationale for this gap is due to selecting the closest-answer to the one arrived in the CoT. However, this heuristic was not very common, especially among reasoning models. We observed this behavior ∼23 % of the time when a reasoning model was correct in QMC-CoT and wrong in Q-CoT (see Table 6 for a breakdown for each model). Furthermore, even when we boosted Q-CoT's performance with the benefit of random guessing, denoted Q-CoT+$k$, nearly every model outperformed on QMC-CoT. For 4 options, Q-CoT+$k$ = score(Q-CoT) $\times 0.75 + 0.25$.

Figure 2 ranks models by their ability to exploit, showing that reasoning models are, in general, better test exploiters. Interestingly, parameter size is not correlated with exploitation among reasoning models. In fact, other than DeepSeek R1 (7B), the most exploitative reasoning models have fewer than 32 B parameters, and the top 3 are smaller than 14 B. In part, this is due to saturation of the QMC-CoT format; nearly all reasoning models attain greater than 90 % accuracy on QMC-CoT so the performance gains by the bigger reasoning models appear in the Q-CoT format. This is especially true for o3, where achieving 77.34 % on Q-CoT makes it hard to diagnose its exploitation. And in part, this is due to DeepSeek R1 (70B) having lower accuracy on both QMC-CoT and Q-CoT than the top reasoning models, suggesting that Qwen models are a better base for RL fine-tuning than Llama, matching recent results by Shao et al. (2025).

### 4.1. Pathways of Exploitation

This section examines the information signals models are using to exploit. We start by analyzing the performance of all models on MC-CoT to quantify how much exploitation is coming from reasoning over the options alone. We then quantify the residual exploitation that arises from leveraging extra information in the question by comparing LLM performance on QMC-CoT and Q-CoT-MC-1T.

**MC-only Exploitation** Figure 8 (see Appendix E) quantifies the ability of each LLM to exploit only the information contained in the options, without seeing the question, to beat random guessing, plotting the accuracy above random guessing for each model on MC-CoT. While most models perform better than random guessing, the reasoning model with the lowest MC-CoT performance is higher than the highest non-reasoning model's performance. Among reasoning models, we observed that the Qwen3 models are the best MC-only exploiters, with Qwen3 (32B) obtaining 13 points above random guessing. In Figure 9, we break down the performance above random guessing each model obtains for each dataset. In general, the most exploitable datasets were the ones that were initially instantiated as MCQA. In fact, ARC, HellaSwag, and PIQA were the datasets most susceptible to MC-only exploitation, with every model attaining a statistically significant accuracy above random guessing, and with all but one reasoning model obtaining higher than 80 % accuracy on PIQA.

**QMC-based Exploitation** We then analyzed the residual exploitation that occurs when LLMs are given the question text along with the options. To do this, we evaluated LLMs when they could reason over the question and the options vs when they could only reason about the question. Here, we ran LLMs on our two-stage configurations; if an LLM's performance on Q-CoT-MC-1T is worse than on QMC-CoT—

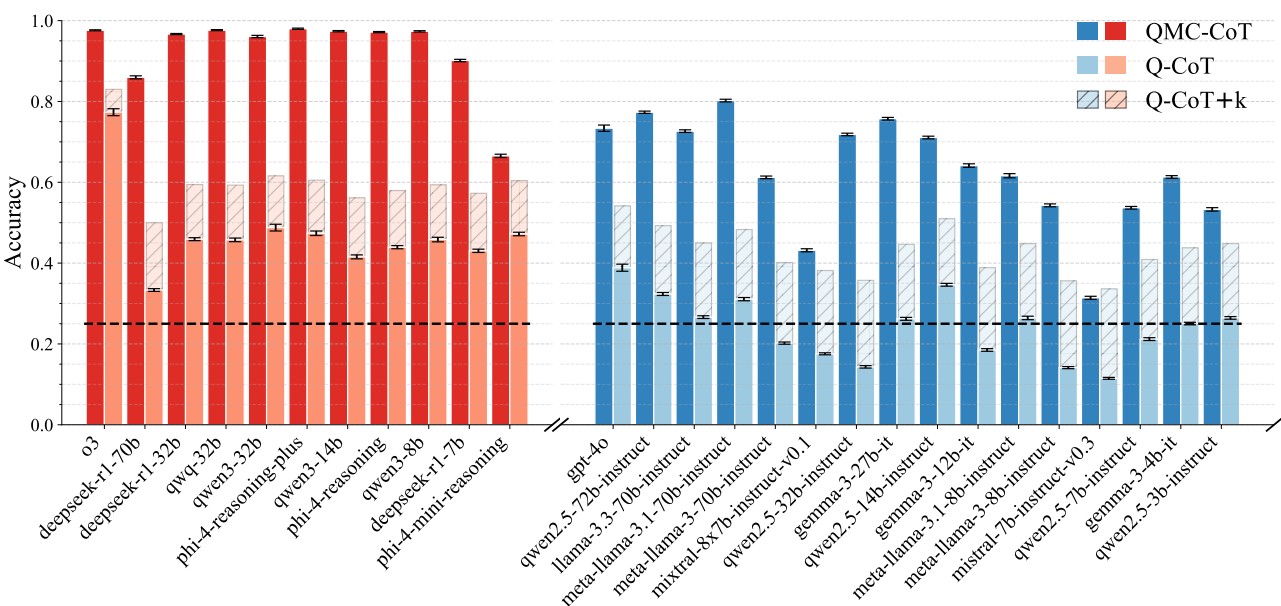

*Figure 1.* Pass@1 accuracy of each LLM on the set of CoT-extractable questions in the benchmark suite over QMC-CoT (dark) and Q-CoT (light). LLMs are grouped into reasoning models (red) and non-reasoning models (blue), sorted by parameter count. Beneath every Q-CoT bar, we plot the boost in accuracy Q-CoT would have gotten with random guessing denoted Q-CoT+$k$.

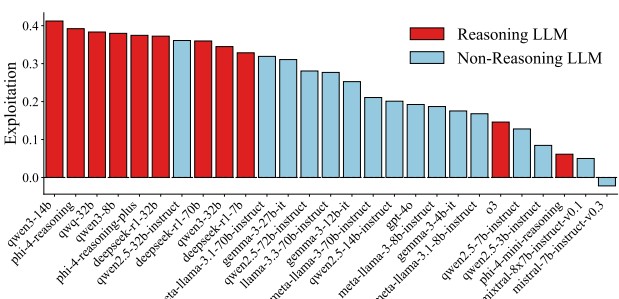

*Figure 2.* The amount of exploitation by each LLM on the set of CoT-extractable questions in the benchmark suite. Reasoning models are in red and non-reasoning models in blue.

corrected by their MC-only exploitation—that would be evidence of QMC-based exploiting behavior. We correct for MC-only exploitation by subtracting a model's QMC-CoT performance by their MC-CoT performance, and their Q-CoT-MC-1T performance by random guessing. To account for any drop in performance due to mapping issues, we super-scored Q-CoT-MC-1T with Q-CoT: if a model was correct on a question on either format then they were deemed correct. Therefore, we define QMC-based exploitation as: $E_{\text{QMC}} = (A_{\text{QMC-CoT}} - A_{\text{MC-CoT}}) - (A_{\text{S}} - A_{\text{MC-1T}})$, where $A_{\text{S}}$ is the super-scored accuracy.

Of course, we cannot evaluate a standard reasoning model on Q-CoT-MC-1T. However, Qwen3 models have the functionality to switch off their reasoning capabilities, allowing us to evaluate them on Q-CoT-MC-1T and compare

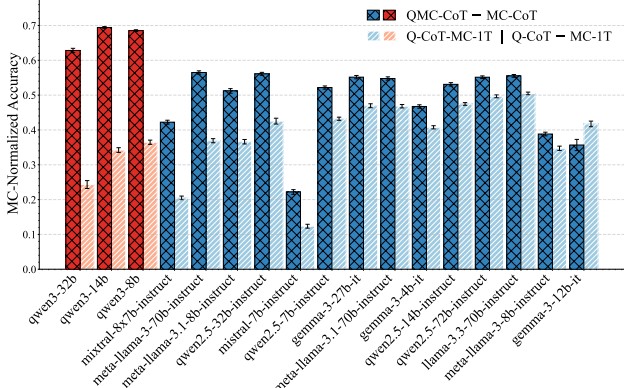

*Figure 3.* The MC normalized accuracy of non-reasoning models (Qwen3 models) on QMC-CoT in dark blue (dark red) and non-reasoning models (Qwen3 thinking mode off in the second step) super-scored on Q-CoT and Q-CoT-MC-1T in light blue (light red). LLMs are sorted by $E_{\text{QMC}}$.

them with non-reasoning models. Figure 3 plots the MC-normalized accuracy for non-reasoning and Qwen models sorted by $E_{\text{QMC}}$. We see widespread evidence of QMC-based exploitation. In fact, all but one LLM exhibited positive $E_{\text{QMC}}$. Furthermore, Qwen models exhibited a greater prevalence of QMC-based exploitation, with larger $E_{\text{QMC}}$ than any non-reasoning model.

## 4.2. Effect of Option Design on Exploitability

Given that LLMs can do better than random guessing just by looking at the options, we asked how specific option sets permit exploitation. We first revisited our MC-only and QMC-based probes to quantify the importance of the presence of the correct answer. Then we compared MMLU to MMLU-Pro to assess the efficacy of "harder" distractors on exploitability.

**Effect of NOTA**  Under MCNA-CoT, the performance above random guessing decreased significantly (see Figure 10 and Figure 11 in the appendix). While ARC, HellaSwag, and PIQA remained highly exploitable datasets, performance on other datasets more closely matched random guessing. As a result, this reduced reasoning models' advantage, where on MC-CoT reasoning models scored $12.63\%$ higher than non-reasoning models but on MCNA-CoT, reasoning models only scored $5.29\%$ higher than non-reasoning models. In part, this is due to higher NOTA selection rates for reasoning models. On average, reasoning models selected NOTA $55.82\%$ of the time as compared to $30.05\%$ by non-reasoning models (the true rate is $25\%$). Inspecting the CoT's, it seems that reasoning models more often considered the MCNA-CoT setting to be a trick question, and NOTA a common answer to trick questions.

We then examined how NOTA affects QMC-based exploitation. We previously observed that Q-CoT-MC-CoT allows reasoning models to refine their answers by re-examining the options, we observed that Q-CoT-MCNA-CoT can disrupt this second-pass shortcut (see Figure 12). Most models exhibited at least some downward shift; suggesting that while these LLMs achieve high accuracy when they can reason over the full option set, their performance drops by 2 to 15 percentage points without the correct answer.

Given the behavior in MCNA-CoT, we test whether performance drops are because NOTA is an attractive distractor or because the correct answer is important for QMC-based exploitation. We treat NOTA selection as a binary classification task and report precision and recall for both classes (Table 8). For questions where NOTA replaces the true answer, DeepSeek R1 (70B) attains precision of $0.85$ and recall of $0.58$. For questions where NOTA is *not* the right answer, precision is $0.78$ and recall is $0.94$, indicating it rarely over-selects NOTA when a correct option exists. These results suggest that the model is not unduly drawn to NOTA as a salient choice; rather, it applies NOTA selectively when its reasoning trace does not map to another valid option. This pattern follows for most reasoning models.

**Effect of Harder Options**  We next examined whether making the option set "harder" (and larger) reduces MC-only exploitation. MMLU and MMLU-Pro offer a natural

testbed for this question. For each dataset, we compute a normalized exploitation: $(k \times A_{\text{MC-CoT}} - 1)/(k - 1)$, so that 0 means random guessing and 1 means perfect accuracy from the options alone. This puts MMLU ($k = 4$) and MMLU-Pro ($k = 10$) on a common scale independent of the number of options.

Two patterns stand out from Figure 13 (in the Appendix): (1) For nearly all non-reasoning models, while MMLU-Pro is strictly harder to exploit than MMLU, the option sets leak enough signal to beat random guessing—with values in the 5 to $10\%$ range. (2) For reasoning models, while MMLU-Pro is often harder to exploit than MMLU, they are able to exploit MMLU-Pro more easily than non-reasoning models exploit MMLU. Together, these results suggest that as models get better at reasoning, they are better able to exploit the information in the option set and avoid "hard" distractors.

## 4.3. Behavior Diagnosis

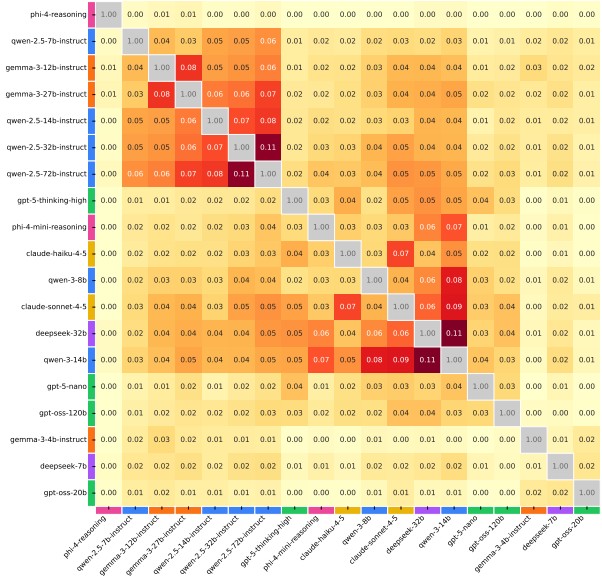

*Figure 4.* The normalized overlap of each LLM's set of unique behavior hashes across all CoT-executable benchmarks. The colors on the x- and y-axis denote model family.

This section studies how often LLMs make the same kinds of mistakes on CoT-executable benchmarks. We take each model's generated code, run it on many controlled input variations, and record the outputs in a compact "behavior signature" by hashing them. If two models have the same signature, they outputted the same logical function, even if their code looks different. This lets us tell when models converge on the same underlying error pattern versus when they fail in fundamentally different ways.

Figure 4 depicts the normalized overlap (Jaccard similarity) of each LLM's set of unique behavior hashes across all incorrect responses. The heatmap reveals two key trends:

(1) LLMs within the same architectural family often exhibit higher correlation in their shared incorrect behaviors than LLMs from different families, suggesting that training data, architecture, or optimization choices propagate similar failure modes within families; and (2) among Qwen-2.5 models specifically, behavioral overlap increases with model size, indicating that as performance increases models in this family converge toward more consistent error patterns. While family clustering is statistically significant, our question-level analysis reveals a more nuanced picture: approximately $75\%$ of behavior groups involve models from different families, indicating that many failure modes are shared across the model landscape.

**Similarity across formats**   To understand whether shared functional errors reflect similar selection errors, we compare behavior hashing with MCQA selection patterns. Since random guessing would produce approximately $33\%$ agreement when both models are wrong, we remove this baseline, yielding similarity above chance. We find a remarkably low correlation (Pearson $r = 0.090$, $p = 0.513$) between these two measures, indicating they capture fundamentally different aspects of model errors. In $54.5\%$ of model pairs, behavior hashing shows higher similarity than MCQA. This pattern is especially pronounced between Qwen and DeepSeek: DeepSeek-R1 (32b) models show strong functional similarity to their source models (e.g., Qwen-2.5 (32b) have Jaccard $0.111$ for behavior hashing but $0.030$ for MCQA), indicating that reasoning patterns are preserved from the base model. Conversely, in $5.5\%$ of pairs, MCQA similarity (avg. $0.606$) far exceeds behavior hashing similarity (avg. $0.040$), suggesting these models make systematic reasoning errors that manifest differently in code implementation.

**Model families cluster at the question-level**   Beyond aggregate similarity patterns, our question-level analysis shows that error clustering can be family-specific. On $18$ problems, the incorrect behavior hashes partition into groups dominated by a single family—for example, one group consisting only of Qwen models and another consisting only of Gemma models—showing that different families can converge on qualitatively different wrong solutions to the same question. This implies that some systematic weaknesses are not universal, but reflect family-specific blind spots. At the same time, within-family convergence on the same functional error (despite different code) indicates these failures are systematic rather than incidental. The following figures illustrate this phenomenon.

MBPP Question 138:

Write a python function to check whether the given number can be represented as sum of non-zero powers of 2 or not.

*Figure 5.* Qwen-2.5 models on MBPP 138. Checks for no consecutive 1s in binary (returns False for numbers like 110, 101,...).

*Figure 6.* Gemma-3 models on MBPP 138. Checks if all bits are 1 (returns True only for numbers like 1, 3, 7...).

## 5. Conclusions

Although LLMs are achieving higher benchmark performances than ever, some of the improvement comes from exploiting the options. Our investigation reveals three lessons for the design and interpretation of LLM evaluations: (1) Decoupling is essential and increasingly infeasible for today's reasoning models. Separating CoT from selection—via `Q-CoT-MC-1T` and, to some extent, `Q-CoT-MCNA-CoT`—can expose latent reasoning ability and distinguish first-principles reasoning from test exploitation. However, this diagnostic relies on the ability to produce *question-only* reasoning before options are shown; which most reasoning models cannot do. As a result, MCQA is no longer a dependable proxy for free-form ability. Moreover, reasoning and selection should be reported separately. (2) Since MCQA is likely here to stay, design for option-independent correctness: write stems that do not reference the options and either define a canonical free-form answer or score via post-hoc mapping. (3) Relying solely on more challenging distractors as an antidote to exploitation is insufficient; while they may increase difficulty, they do not reliably mitigate test exploitation and must be used sparingly.

Ultimately, all we can observe is what we measure. Without careful design, high test performance may reflect proficiency in exploiting the test rather than true competence. As LLMs continue to improve and are used in the real-world, it becomes increasingly important to align what we measure with what we value.

## Acknowledgements

This work was funded by an NSERC Discovery Grant, a CIFAR Canada AI Research Chair (Alberta Machine In-

telligence Institute), and computational resources provided both by UBC Advanced Research Computing and a Digital Research Alliance of Canada RAC Allocation.

## Impact Statement

This paper studies how multiple-choice evaluations can give large language models an opportunity to appear more capable than they are on open-ended reasoning tasks. Our findings are relevant to researchers, companies, and policymakers who use benchmark scores to compare models or make deployment decisions. If multiple-choice scores are treated as direct evidence of reasoning ability, users may overestimate model reliability in settings where no answer options are available. This could lead to misplaced trust in AI systems, especially in domains where errors are costly. By identifying when and how models exploit answer choices, our work encourages more careful evaluation practices: reporting open-ended and multiple-choice performance separately, designing questions whose correctness does not depend on the option set, and using richer diagnostics when analyzing model failures.

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

# A. Prompts

**User Message:**

Q: <Question text here>

A.  <Option A>

B.  <Option B>

C.  <Option C>

D.  <Option D>

Please reason step by step, and put your final answer within \boxed{}

*Prompt 1:* This is for the Q-CoT configuration.

**User Message:**

Q: <Question text here>

A.  <Option A>

B.  <Option B>

C.  <Option C>

D.  <Option D>

Answer by writing the option letter corresponding to the correct option. WRITE ONLY A SINGLE LETTER.
A:

*Prompt 2:* This is for the QMC-CoT configuration.

The other configurations either entirely omit the "Q: <Question text here>" (MC-CoT) or, in the case of the two-stage configurations, first prompt with Q-CoT and then prompt with QMC-CoT but omit the "Q: <Question text here>."

**User Message:**

Solve the following problem. Write Python code as general as possible that computes the answer. I will be running your code with other inputs, so make sure it is correct and general. Define a function `solve` that takes the numeric values as parameters and returns the answer. Print the result.
**Rules:**

- Keep the code runnable with standard library only.

- Make sure the function returns the correct answer.

—

**Example:**
Q: A store has 12 apples. They sell 5. How many apples remain?

```python
def solve(total, sold):
    remaining = total - sold
    return remaining

result = solve(12, 5)
print(result)
```

—

**Your Problem:**

*Prompt 3:* This is for the executable CoT diagnostic configuration.

# B. Models

| Model Name | Model Card | Reasoning |
|---|---|---|
| **Closed-Weight** | | |
| *OpenAI* | | |
| gpt-5 | https://openai.com/index/introducing-gpt-5/ | ✓ |
| gpt-5-mini | https://openai.com/index/introducing-gpt-5/ | ✓ |
| o3 | https://openai.com/index/o3-o4-mini-system-card/ | ✓ |
| GPT-4o | https://openai.com/index/gpt-4o-system-card/ | × |
| *Anthropic* | | |
| claude-sonnet-4-5 | https://www.anthropic.com/news/claude-sonnet-4-5 | ✓ |
| claude-haiku-4-5 | https://www.anthropic.com/news/claude-haiku-4-5 | ✓ |
| **Open-Weight** | | |
| *DeepSeek* | | |
| DeepSeek-R1-70B | https://huggingface.co/deepseek-ai/DeepSeek-R1-Distill-Llama-70B | ✓ |
| DeepSeek-R1-32B | https://huggingface.co/deepseek-ai/DeepSeek-R1-Distill-Qwen-32B | ✓ |
| DeepSeek-R1-7B | https://huggingface.co/deepseek-ai/DeepSeek-R1-Distill-Qwen-7B | ✓ |
| *Microsoft* | | |
| Phi-4-reasoning-plus | https://huggingface.co/microsoft/Phi-4-reasoning-plus | ✓ |
| Phi-4-reasoning | https://huggingface.co/microsoft/Phi-4-reasoning | ✓ |
| Phi-4-mini-reasoning | https://huggingface.co/microsoft/Phi-4-mini-reasoning | ✓ |
| *Qwen* | | |
| Qwen2.5-72B-Instruct | https://huggingface.co/Qwen/Qwen2.5-72B-Instruct | × |
| Qwen2.5-32B-Instruct | https://huggingface.co/Qwen/Qwen2.5-32B-Instruct | × |
| Qwen2.5-14B-Instruct | https://huggingface.co/Qwen/Qwen2.5-32B-Instruct | × |
| Qwen2.5-7B-Instruct | https://huggingface.co/Qwen/Qwen2.5-7B-Instruct | × |
| Qwen3-32B | https://huggingface.co/Qwen/Qwen3-32B | ✓ |
| Qwen3-14B | https://huggingface.co/Qwen/Qwen3-14B | ✓ |
| Qwen3-8B | https://huggingface.co/Qwen/Qwen3-8B | ✓ |
| *Google* | | |
| gemma-3-27b-it | https://huggingface.co/google/gemma-3-27b-it | × |
| gemma-3-12b-it | https://huggingface.co/google/gemma-3-12b-it | × |
| gemma-3-4b-it | https://huggingface.co/google/gemma-3-4b-it | × |
| *Meta Llama* | | |
| Llama-3-8B-Instruct | https://huggingface.co/meta-llama/Meta-Llama-3-8B-Instruct | × |
| Llama-3-70B-Instruct | https://huggingface.co/meta-llama/Meta-Llama-3-70B-Instruct | × |
| Llama-3.1-8B-Instruct | https://huggingface.co/meta-llama/Meta-Llama-3.1-8B-Instruct | × |
| Llama-3.1-70B-Instruct | https://huggingface.co/meta-llama/Meta-Llama-3.1-70B-Instruct | × |
| Llama-3.3-70B-Instruct | https://huggingface.co/meta-llama/Llama-3.3-70B-Instruct | × |
| *Mistral* | | |

| Mistral-7B-Instruct-v0.3 | https://huggingface.co/mistralai/Mistral-7B-Instruct-v0.3 | × |

*Table 5.* Overview of the open- and closed-weight LLMs we evaluated. The table includes their names, their model card links, and whether they have been chat or instruction tuned. Models are grouped by family and sorted by parameter size, with non-chat-tuned models listed first within each group.

## C. Dataset Conversion and Methods

### C.1. Answer Extraction

```python
def evaluate_anwer(ma, ca):
    ma = model_answer.strip()
    ca = correct_answer.strip()

    def numeric_comparison(ma, ca):
        mf = float(ma)
        cf = float(ca)
        # digits after decimal in model float
        s = str(mf)
        sig = len(s.split('.')[1]) if '.' in s else 0
        return mf == round(cf, sig)

    def get_numeric_value(s):
        nums = re.findall(r"[-+]?(?:\d*\.\d+|\d+)", s)
        return [float(n) if "." in n else int(n) for n in nums]

    # 1) Try pure numeric comparison with sig figs
    try:
        return numeric_comparison(ma, ca)
    except ValueError:
        # If it fails, it means the model answer is not a number
        pass

    # 2) Try to canonicalize common LaTeX into Python/SymPy
    try:
        ma_py = _latex_to_python(ma)
        ca_py = _latex_to_python(ca)
        expr_ma = parse_expr(ma_py, transformations=_transformations)
        expr_ca = parse_expr(ca_py, transformations=_transformations)
        # True if their difference simplifies to 0
        return simplify(expr_ma - expr_ca) == 0
    except Exception:
        return None
```

## C.2. MMLU

The programmatic filtering we used:

```python
import re
from string import ascii_lowercase

# catch "Which of the following", "Select the", "Choose", "All of the following except"
MCQ_KW = re.compile(
    r'\b(?:which of the following|select the|all of the following except|which one of the following|which
    ↪  statement|which sequence|which of one of the following|which is the most|which will most
    ↪  likely|which process|what can be concluded from the passage|_)\b',
    flags=re.IGNORECASE
)

def needs_options_by_keyword(q: str) -> bool:
    return bool(MCQ_KW.search(q))

def has_open_ended_ending(q: str) -> bool:
    return q.strip()[-1].lower() in ascii_lowercase

def has_duplicate_options(row):
    # if any two options are the same, remove the question
    option_set = set([elm['text'] for elm in row['options']])
    if len(option_set) != len(row['options']):
        return True
    return False
```

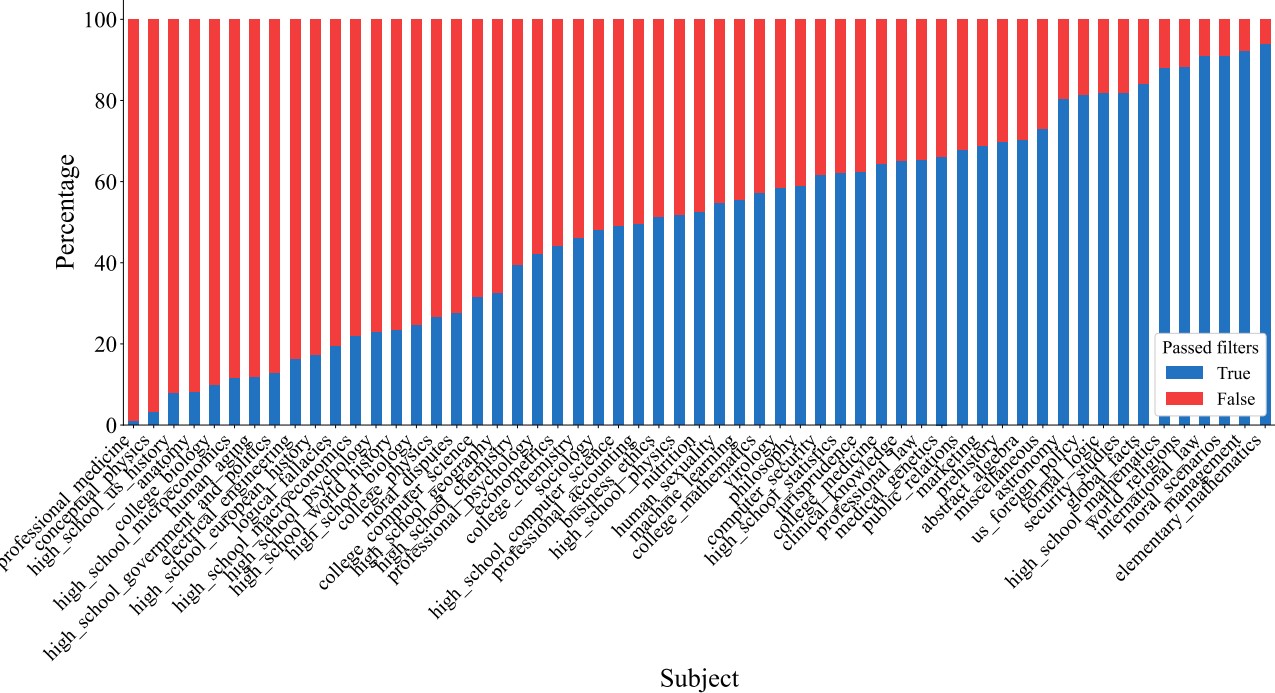

*Figure 7.* This figure plots the percentage of questions (by subject) that passed the filters we ran on the MMLU portion of the Open-LLM benchmark. We note that there was not a systematic removal of "reasoning" subjects over answer retrieval subjects.

# D. Behavior probing

Numeric values are probed uniformly in the range $[-1000, 1000]$, with integers used for whole numbers and floats for fractional values. For duplicate argument values (e.g., solve(5, 5, 12)), we generate all permutations of tie assignments

to handle semantic ambiguity, producing multiple behavior hashes per sample when ties exist.

# E Figures

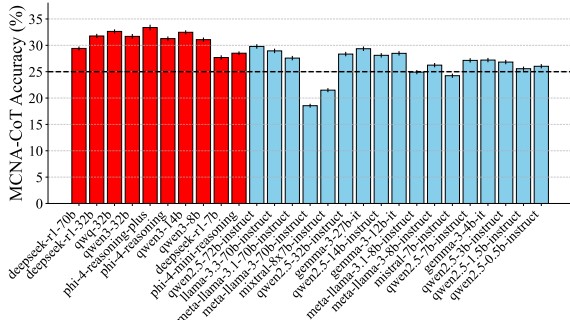

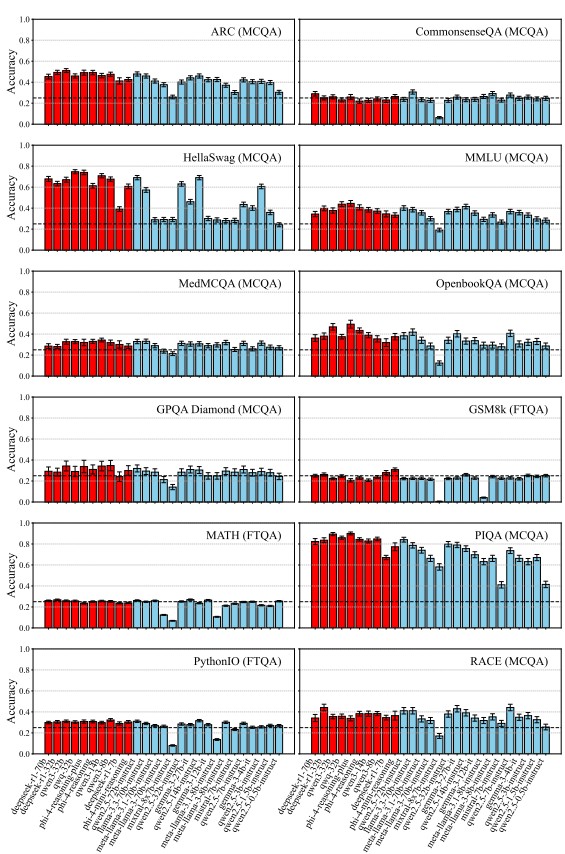

*Figure 8.* This figure plots the accuracy for each LLM on MC-CoT. In red are reasoning models and in blue are the non-reasoning models. The black line is the accuracy random guessing achieves. Note that some non-reasoning models perform *worse* than random guessing; they were systematically biased by signals in the options that were correlated against the correct answer.

*Figure 10.* This figure plots the accuracy for each LLM on MCNA-CoT. Random guessing is the black line, in red are reasoning models, in blue are the non-reasoning models, models are sorted by parameter. We see that all models achieve closer to random-guessing performance, even those below-chance, implying that inclusion of NOTA also diminishes the ability to identify spurious signals in the options.

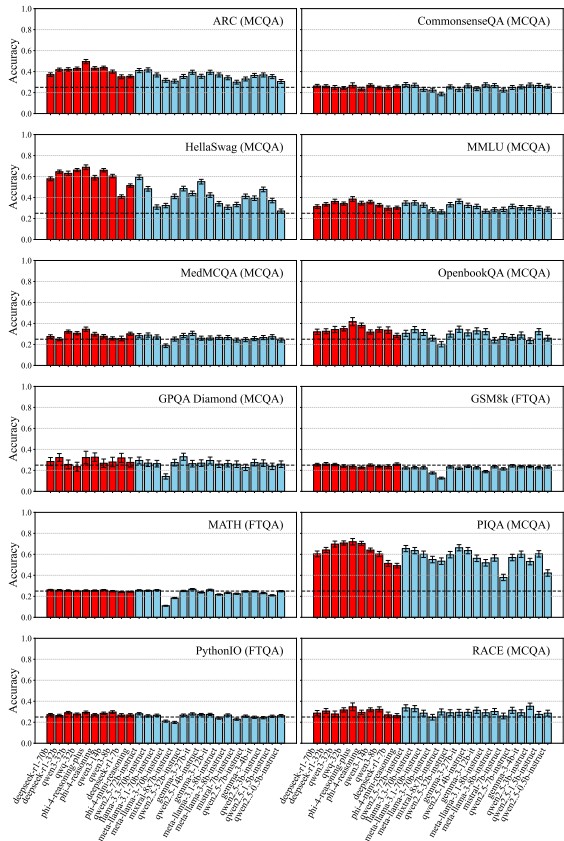

*Figure 9.* This figure plots the accuracy for each LLM on MC-CoT on each dataset. In red are reasoning models and in blue are the non-reasoning models. Within each group, models are sorted by parameter. We see two general trends in this figure: (1) Many MCQA benchmarks contain enough information in the options alone for most models to beat random guessing, and (2) the datasets that induce lower than random guessing are usually FTQA datasets with generated options.

*Figure 11.* This figure plots the accuracy for each LLM on MC-CoT on each dataset. In red are reasoning models and in blue are the non-reasoning models. Within each group, models are sorted by parameter. We see similar trends as above, LLMs perform closer to random guessing, decreasing above-chance performance and increasing below-chance performance. Furthermore, MCQA benchmarks still remain more exploitable albeit less so than on MC-CoT.

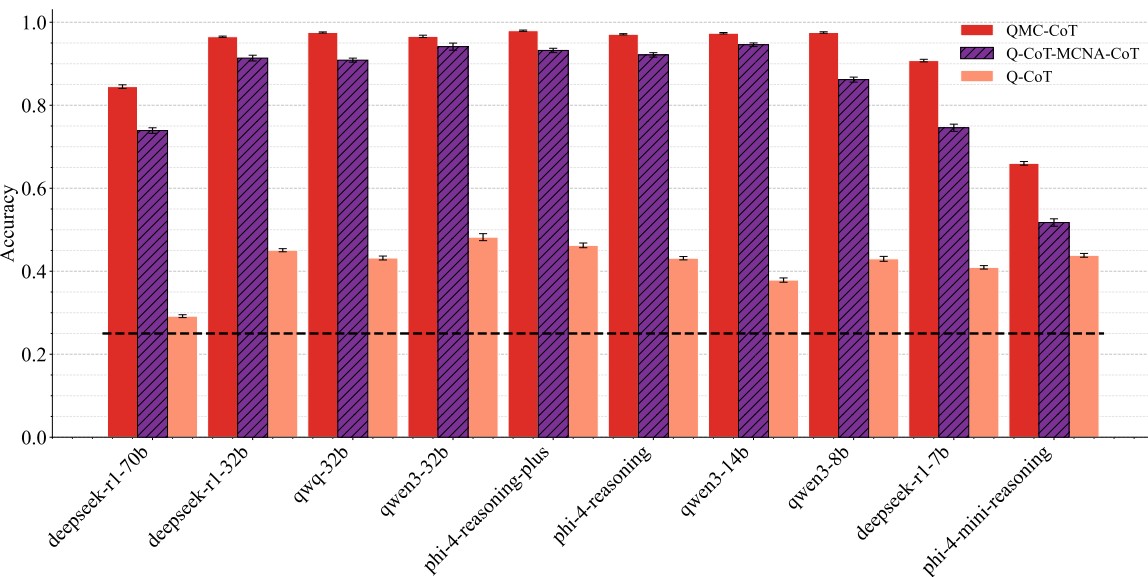

*Figure 12.* This figure plots the accuracies of all reasoning models on QMC-CoT (dark red), Q-CoT-MCNA-CoT (purple), and Q-CoT (light red), sorted by parameter size. The dotted black line indicates the accuracy random guessing achieves. We see that every LLM's accuracy decreases when evaluated on Q-CoT-MCNA-CoT from QMC-CoT, with the smaller LLMs seeing larger performance drops. This suggests that larger LLMs are more capable of exploiting the options even without the correct answer.

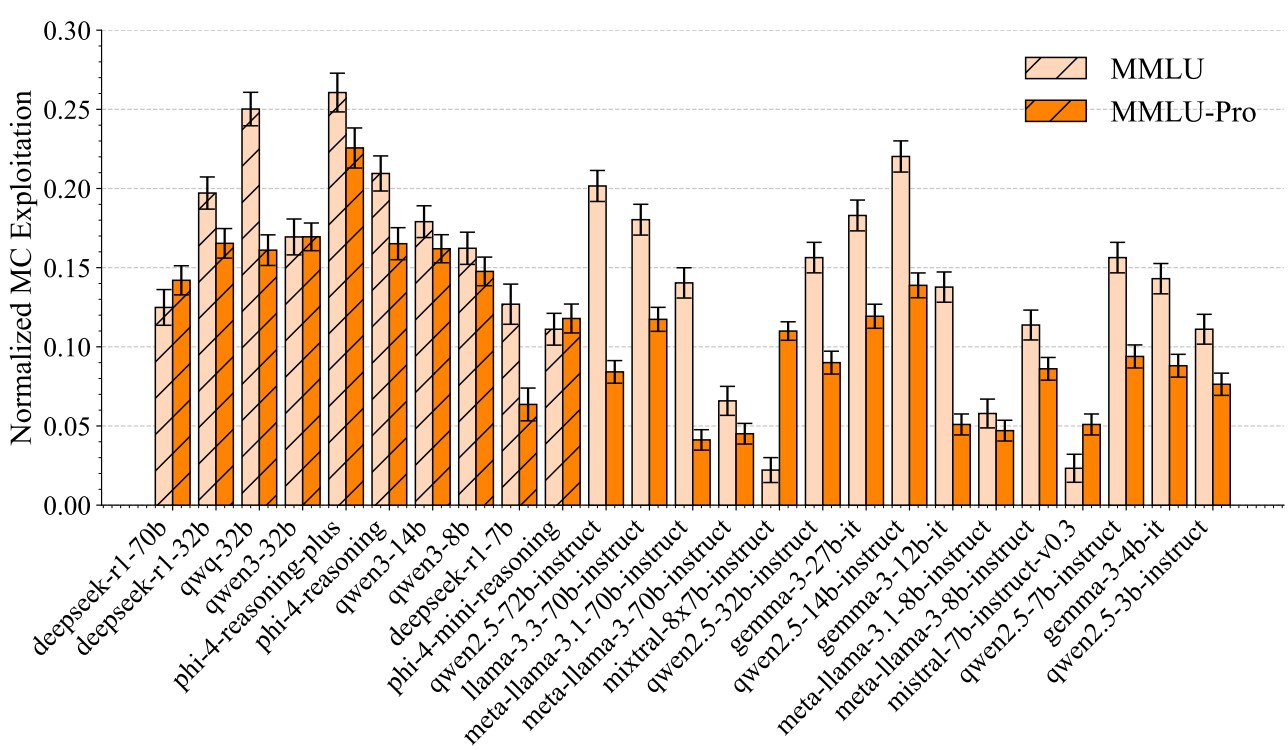

*Figure 13.* The normalized MC-only exploitation of all models on `MMLU` and `MMLU-Pro`. Reasoning models are hatched.

## F. Tables

| Model | (%) |
|---|---|
| DeepSeek-R1-Llama-70B | 20.80 |
| DeepSeek-R1-Qwen-32B | 27.78 |
| DeepSeek-R1-Qwen-7B | 22.99 |
| Meta-Llama-3-70B-Instruct | 23.60 |
| Meta-Llama-3-8B-Instruct | 26.02 |
| Meta-Llama-3.1-8B-Instruct | 44.23 |
| Mixtral-8x7B-Instruct-v0.1 | 34.81 |
| Mistral-7B-Instruct-v0.3 | 54.48 |
| Phi-4-reasoning-plus | 20.69 |
| Phi-4-reasoning | 30.61 |
| Qwen2.5-72B-Instruct | 21.28 |
| Qwen2.5-32B-Instruct | 33.33 |
| Qwen2.5-14B-Instruct | 18.42 |
| Qwen2.5-7B-Instruct | 57.38 |
| Qwen2.5-3B-Instruct | 29.85 |
| Gemma-3-27b-it | 59.02 |
| Gemma-3-12b-it | 85.71 |
| Gemma-3-4b-it | 82.22 |

*Table 6.* This table depicts the percent of the time an LLM chooses the correct answer in QMC-CoT due to selecting the closest answer they derived in their Q-CoT response (which was incorrect).

| Model | ✓on Q-CoT, × on Q-CoT-MC-1T |
|---|---|
| DeepSeek-R1-Llama-70B | 3.97 % |
| DeepSeek-R1-Qwen-32B | 1.91 % |
| DeepSeek-R1-Qwen-7B | 18.25 % |
| Meta-Llama-3-70B-Instruct | 9.30 % |
| Meta-Llama-3-8B-Instruct | 0.78 % |
| Meta-Llama-3.1-8B-Instruct | 15.20 % |
| Mixtral-8x7B-Instruct-v0.1 | 27.41 % |
| Mistral-7B-Instruct-v0.3 | 8.87 % |
| Phi-4-reasoning-plus | 56.31 % |
| Phi-4-reasoning | 55.32 % |
| Qwen2.5-72B-Instruct | 40.00 % |
| Qwen2.5-32B-Instruct | 67.50 % |
| Qwen2.5-14B-Instruct | 63.25 % |
| Qwen2.5-7B-Instruct | 51.80 % |
| Qwen2.5-3B-Instruct | 44.03 % |
| Gemma-3-27b-it | 41.69 % |
| Gemma-3-12b-it | 31.44 % |
| Gemma-3-4b-it | 44.65 % |

*Table 7.* This table lists the percent of the time that models are correct in Q-CoT but then select the wrong answer in Q-CoT-MC-1T.

| Class | precision | recall | f1-score |
|---|---|---|---|
| NOTA incorrect | 0.78 | 0.94 | 0.85 |
| NOTA correct | 0.85 | 0.58 | 0.69 |

*(a) DeepSeek-R1-Distill-Llama-70B*

| Class | precision | recall | f1-score |
|---|---|---|---|
| NOTA incorrect | 0.79 | 0.92 | 0.85 |
| NOTA correct | 0.82 | 0.60 | 0.69 |

*(b) DeepSeek-R1-Distill-Qwen-32B*

| Class | precision | recall | f1-score |
|---|---|---|---|
| NOTA incorrect | 0.72 | 0.88 | 0.79 |
| NOTA correct | 0.71 | 0.44 | 0.55 |

*(c) DeepSeek-R1-Distill-Qwen-7B*

| Class | precision | recall | f1-score |
|---|---|---|---|
| NOTA incorrect | 0.74 | 0.90 | 0.82 |
| NOTA correct | 0.73 | 0.46 | 0.56 |

*(d) Meta-Llama-3-70B-Instruct*

| Class | precision | recall | f1-score |
|---|---|---|---|
| NOTA incorrect | 0.67 | 0.84 | 0.74 |
| NOTA correct | 0.60 | 0.37 | 0.46 |

*(e) Meta-Llama-3-8B-Instruct*

| Class | precision | recall | f1-score |
|---|---|---|---|
| NOTA incorrect | 0.66 | 0.89 | 0.76 |
| NOTA correct | 0.67 | 0.34 | 0.45 |

*(f) Meta-Llama-3.1-8B-Instruct*

| Class | precision | recall | f1-score |
|---|---|---|---|
| NOTA incorrect | 0.67 | 0.84 | 0.74 |
| NOTA correct | 0.60 | 0.37 | 0.46 |

*(g) Meta-Llama-3-8B-Instruct*

| Class | precision | recall | f1-score |
|---|---|---|---|
| NOTA incorrect | 0.66 | 0.89 | 0.76 |
| NOTA correct | 0.67 | 0.34 | 0.45 |

*(h) Meta-Llama-3.1-8B-Instruct*

| Class | precision | recall | f1-score |
|---|---|---|---|
| NOTA incorrect | 0.66 | 0.79 | 0.72 |
| NOTA correct | 0.49 | 0.32 | 0.39 |

*(i) Mistral-7B-Instruct-v0.3*

| Class | precision | recall | f1-score |
|---|---|---|---|
| NOTA incorrect | 0.69 | 0.84 | 0.76 |
| NOTA correct | 0.57 | 0.36 | 0.44 |

*(j) Mixtral-8x7B-Instruct-v0.1*

| Class | precision | recall | f1-score |
|---|---|---|---|
| NOTA incorrect | 0.66 | 0.76 | 0.71 |
| NOTA correct | 0.41 | 0.31 | 0.35 |

*(k) Phi-4-mini-reasoning*

| Class | precision | recall | f1-score |
|---|---|---|---|
| NOTA incorrect | 0.01 | 1.00 | 0.02 |
| NOTA correct | 1.00 | 0.14 | 0.24 |

*(l) Phi-4-reasoning*

| Class | precision | recall | f1-score |
|---|---|---|---|
| NOTA incorrect | 0.02 | 1.00 | 0.04 |
| NOTA correct | 1.00 | 0.12 | 0.22 |

*(m) Phi-4-reasoning-plus*

| Class | precision | recall | f1-score |
|---|---|---|---|
| NOTA incorrect | 0.78 | 0.63 | 0.69 |
| NOTA correct | 0.21 | 0.36 | 0.27 |

*(n) QwQ-32B*

| Class | precision | recall | f1-score |
|---|---|---|---|
| NOTA incorrect | 0.66 | 0.57 | 0.61 |
| NOTA correct | 0.38 | 0.48 | 0.42 |

*(o) Qwen2.5-72B-Instruct*

| Class | precision | recall | f1-score |
|---|---|---|---|
| NOTA incorrect | 0.41 | 0.27 | 0.33 |
| NOTA correct | 0.16 | 0.27 | 0.20 |

*(p) Qwen2.5-32B-Instruct*

| Class | precision | recall | f1-score |
|---|---|---|---|
| NOTA incorrect | 0.67 | 0.77 | 0.72 |
| NOTA correct | 0.39 | 0.27 | 0.32 |

*(q) gemma-3-27b-it*

| Class | precision | recall | f1-score |
|---|---|---|---|
| NOTA incorrect | 0.64 | 0.81 | 0.72 |
| NOTA correct | 0.46 | 0.26 | 0.33 |

*(r) gemma-3-12b-it*

*Table 8.* Classification metrics (precision, recall, F1) for each model on NOTA-incorrect vs. NOTA-correct.

