# OpenReview forum: "Reasoning Models Are Test Exploiters: Rethinking Multiple Choice"
_ICML.cc/2026/Conference — ICML 2026 regular_

### Official Review · Reviewer_pKnV · 2026-03-01

**Soundness:** 3
**Presentation:** 3
**Significance:** 4
**Originality:** 3
**Overall Recommendation:** 5
**Confidence:** 4

**Summary:**

This paper argues that mcqa increasingly overestimates llm's reasoning ability because models exploit answer options rather than rely solely on underlying knowledge. The authors test 27 models and 15 benchmarks and mainly show that reasoning models gain substantial performance boosts when options are visible and harder distractors do not reliably eliminate this effect. When comparing the reasoning traces of models they further demonstrate that shared multiple-choice errors do not reliably reflect shared reasoning processes.

**Compliance With Llm Reviewing Policy:**

Affirmed.

**Key Questions For Authors:**

Please refer to the weaknesses.

**Limitations:**

The authors do not provide an impact statement section.

**Strengths And Weaknesses:**

**Strengths**

This paper addresses an important question about whether mcqa evaluation reliably measures reasoning in modern LLMs. Its methodology is careful, the experiments are broad and systematically designed across many models and benchmarks. The behavioral hashing analysis of reasoning traces is technically interesting. Overall the paper provides measured key findings.

**Weaknesses**

- The NOTA intervention is interesting but the higher NOTA selection rates for reasoning models might have alternative explanations beyond exploitation. For example these models may be more cautious using NOTA as a safe fallback rather than exploiting the option structure itself. It would be helpful to further analyze uncertainty to better separate strategic exploitation from conservative answer selection behavior.

- The paper would also benefit from a limitations or broader impact discussion/section.

_Comment/suggestion:_ The experimental design is thorough but the large number of configurations and abbreviations (QMC-CoT, Q-CoT-MC-1T, MCNA-CoT etc.) can make parts of the results hard to follow. A clearer visual overview (beyond or replacing Table 1) connecting the different setups to the corresponding metrics would make the paper easier to read.

---

> ### Author Rebuttal · Authors · 2026-03-27
>
> Thanks for the thoughtful review! On the NOTA point, we agree that higher NOTA selection rates by reasoning models do not by themselves uniquely identify exploitation. Our intended claim is narrower: NOTA reduces MCQA performance, and the observed behavior is consistent with selective use of NOTA rather than being unduly attracted to it. The current draft includes an analysis treating NOTA selection as a binary classification task: when NOTA is not the correct answer, reasoning models typically do not over-select it, and for DeepSeek-R1 (70B) specifically we report precision (0.78) and recall (0.94) on the non-NOTA class, alongside precision (0.85) and recall (0.58) when NOTA is the correct answer. We therefore view the evidence as suggesting that these models are not simply using NOTA as a generic safe fallback. We also agree that getting at some baser level of uncertainty analysis might be very useful in identifying why models are selecting NOTA on a particular question. Our analysis works in an aggregate manner, but better uncertainty measures could allow us to talk about option selection at a more fine-grained question level.
>
> We also agree that the paper would benefit from a broader impact discussion. We will add one. In particular, we will surface limitations that are currently only implicit in the draft: we focus on inflationary effects rather than cases where MCQA may deflate scores; our question conversion keeps only the subset of questions that are viable standalone free-form items; and for most reasoning models the fully decoupled two-stage setting is not realizable, so some of our two-stage configurations are best interpreted as diagnostics of option sensitivity rather than pure reasoning measures.

---

> > ### Author Rebuttal · Reviewer_pKnV · 2026-04-02
> >
> > Thank you for the response. I will keep my score.

---

### Official Review · Reviewer_i77i · 2026-03-08

**Soundness:** 3
**Presentation:** 3
**Significance:** 2
**Originality:** 2
**Overall Recommendation:** 4
**Confidence:** 4

**Summary:**

This paper raises the issue of LLMs “shortcutting” on multiple-choice questions by exploiting the options' format and content, significantly boosting their performance. The authors argue that MCQA can no longer be used as a good proxy for free-form answering and knowledge evaluation, especially for reasoning models. To support this claim, they design diagnostic probes for isolating option- and question-plus-option answer effect-exploitability and perform experiments with them on multiple modern LLMs and several MCQA datasets.

**Compliance With Llm Reviewing Policy:**

Affirmed.

**Final Justification:**

Authors' response addresses most of the issues raised by me. Although, I still have some concerns about the degree of novelty, I think that the incorporation of the proposed changes and the proposed adjustment in positioning will greatly benefit the article and I raise my overall score to 4.

**Key Questions For Authors:**

- In two-stage reasoning, do you prompt the model (at first step) with the fact that it will be given list of options later?
- Would expanding the option list with “I don’t know” help address the issue of random guessing? and what overall effect it would cause?
- Can you please explain the intuition behind the formula for E in line 216-right?
- Please update reference to `Balepur, N., Ravichander, A., and Rudinger, R. Artifacts or abduction: How do llms answer multiplechoice questions without the question?` from  arXiv preprint to the  proper ACL paper reference (doi `10.18653/v1/2024.acl-long.555` ) . The same issue applies to a number of other sources in bibliography.

**Limitations:**

No, limitations are not discussed. Overall this is a fundamental work, so there is no potential negative impact can be directly identified.

**Strengths And Weaknesses:**

Strengths :
- An extensive coverage of various scenarios scenarios of coupling MCQA with chain-of-thought reasoning.. The experiments are performed on a large number of modern LLMs from different families.

- This paper studies a rarely covered setting of CoT on executable benchmarks and "code-expressed" reasoning and introduces an interesting approach of "behaviour signaturing".
- In general, this paper is well-written and easy to follow.

Weaknesses:

My main concern with this paper is the lack of novelty of this topic and presented findings. MCQA-format questions are used for knowledge assessment in various educational systems throughout the world, and their validity as a proxy for this role has received a lot of criticism exactly for the reasons outlined in the conclusion of this paper (random guessing/option exploitation/etc.). Similar issues for LLMs are also quite well-known and raised in articles cited by this paper. [1] (although it is quite recent) analyses the reasoning traces produced by LLMs when they perform “test time exploitation” and identifies reasoning patterns quite similar to those exhibited by human agents when they try to answer quiz questions for which they do not know the correct answer.

Secondly, there is some issue with article positioning. The abstract and contribution list stress the proposal of guidelines to mitigate the identified issue; however, it is actually done in a single paragraph in conclusion, and the proposed tactics aren’t properly analysed.

Also, the experimental setup in QMC-COT vs Q-CoT comparison may be somewhat unfair towards the latter, leading to overestimation of the exploitation gap. Problematic seems to be the case when the question allows for a quite broader range of potentially correct answers but only one of which is included in the options. For instance, there are many such questions in CommonsenseQA. One can also argue that such questions require inference “from the opposite”, that is, essentially test time exploitation.


1. `Nishant Balepur, Atrey Desai, Rachel Rudinger. Test-Time Reasoners Are Strategic Multiple-Choice Test-Takers. ArXiv preprint:2510.07761`

---

> ### Author Rebuttal · Authors · 2026-03-27
>
> Thanks for the thorough and thoughtful review! We agree that the broad concern that multiple-choice questions can be shortcut is not new, either in educational measurement or in recent LLM work. Our intended contribution is therefore not to claim discovery of the phenomenon in general, but to make four more specific contributions for modern LLM evaluation: (i) a decomposition of when excess performance arises by varying CoT timing and option exposure, (ii) the observation that the diagnostically appealing "decoupled" setup is increasingly infeasible for reasoning models because they reason over whatever is present in the prompt, (iii) controlled analyses of option-design interventions such as NOTA and harder distractors, and (iv) a complementary diagnostic beyond option choice using executable CoT traces. Furthermore, thank you for flagging this recent preprint. We agree it is important related work and will cite and discuss it in the revision. We view it as subsequent rather than concurrent as it cites our preprint as well.
>
> Furthermore, while much of the technical contribution of the paper is about measurement and diagnosing failure modes, each of these contributions carries along with it a downstream implication for benchmark design. We will revise the text to make that emphasis clearer. At present, the paper's concrete recommendations are: (i) decouple reasoning from answer selection whenever possible and report them separately, (ii) design for option-independent correctness by writing stems that can stand alone and by defining a free-form answer, and (iii) avoid treating harder distractors as a sufficient fix, since they cannot reliably reduce exploitation. We agree these should be surfaced more prominently.
>
> On the concern that our setup may unfairly disadvantage Q-CoT in some settings, we think this is a great and subtle point. We took several steps to reduce the impact of this: (1) our grading for free-form answers uses equivalence checks rather than literal option matching whenever possible, (2) in the two-stage comparison we super-score Q-CoT-MC-1T with Q-CoT so that if a model is correct in either format the answer is counted as correct, and (3) we explicitly include benchmarks where there is a single correct answer and do our best to filter to benchmarks without these ambiguities. On this third point, we find that there is not much qualitative difference across the benchmarks we evaluated on, but we will make this caveat more explicit.
>
> **Answers:**
> 1. No, we treated it like a free-form response and asked the model to return their answer in \\boxed{}. We only exposed any mention of options in the second step.
> 2. We expect an abstention-like option to mitigate some forms of random guessing, but not to solve the core issue. Our NOTA experiments already point in that direction: NOTA reduces above-chance performance in MC-only settings and can disrupt second-pass shortcutting, but it does not eliminate exploitation. So we would view "I don't know" as a potentially useful calibration device, not a full remedy.
> 3. The basic intuition comes from the fact that MCQA and FTQA are in some sense on different scales: 25\% accuracy on FTQA should be viewed as higher performance than 25\% accuracy on MCQA. So, to measure how different the gap between those two scores are we need to put them on the same scale. One simple way of doing this is realizing that MCQA is boosted by random guessing, so we credit FTQA by this boost ($1/k$) and normalize it so that its values remain between 0 and 1 ($k-1/k$). Therefore, we get the gap $E = A_{MC} - A_{FT}\cdot(k-1)/k+1/k$. We then expand this measure to incorporate different ways that exploitation can occur and credit FTQA such that we can isolate those particular pathways.
> 4. Thanks for pointing this out! We will update all out-of-date references in the revision.

---

> > ### Author Rebuttal · Reviewer_i77i · 2026-04-04
> >
> > Thank you for the provided response. It addresses most of the issues raised by me. Although, I still have some concerns about the degree of novelty, I think that the incorporation of the proposed changes and the proposed adjustment in positioning will greatly benefit the article.

---

### Official Review · Reviewer_iyGF · 2026-03-11

**Soundness:** 4
**Presentation:** 4
**Significance:** 4
**Originality:** 4
**Overall Recommendation:** 6
**Confidence:** 2

**Summary:**

This paper studies how multiple-choice question answering (MCQA) can inflate the measured reasoning ability of large language models (LLMs). The authors evaluate 27 models across 15 benchmarks using formats that vary whether answer options are shown before or after reasoning. They show that models—especially reasoning models—perform dramatically better when options are visible, indicating that they exploit signals in the answer choices rather than solving the problem independently. Even options alone can produce above-random accuracy. Experiments with “none-of-the-above” and harder distractors show that option design only partially mitigates exploitation. The paper concludes that MCQA is a weak proxy for reasoning and recommends separating reasoning from answer selection in benchmark design.

**Compliance With Llm Reviewing Policy:**

Affirmed.

**Final Justification:**

This is a well-written and interesting paper that initially attracted me, leading me to give a high rating. After reading Reviewer i77i’s comments about limited novelty in light of a very recent related work, I realized I am not very familiar with this specific direction. I will therefore decrease my confidence while keeping my rating.

**Key Questions For Authors:**

1. I think some of the multiple-choice benchmarks you used originally have None-of-the-Above as a choice for some questions. Did you filter them out? Or maybe it doesn't matter?
2. Can you provide a takeaway suggestion for benchmark design in practice at the end of the paper or in the Appendix?

**Limitations:**

yes

**Strengths And Weaknesses:**

Strengths:
1. Well-written.
2. Great survey in the introduction.
3. Great experiment design.
4. Insightful analysis.

Great paper overall, no major weakness.

---

> ### Author Rebuttal · Authors · 2026-03-27
>
> Thanks for your thoughtful review!
>
> **Answers:**
> 1. This is a good question, and we should have been more clear in the text. During the filtering process we removed questions which contained options with any NOTA-like text (e.g., "none of the following") so that our reported NOTA results come from a controlled intervention that we introduce ourselves.
> 2. We agree this should be stated more explicitly. We do include three takeaways in the conclusion, but we will add a short benchmark-design checklist emphasizing: (i) decouple reasoning from answer selection whenever possible and report them separately, (ii) design for option-independent correctness by writing stems that can stand alone and by defining a free-form answer, and (iii) avoid treating harder distractors as a sufficient fix, since they cannot reliably reduce exploitation.

---

> > ### Author Rebuttal · Reviewer_iyGF · 2026-03-31
> >
> > Thank you for your thoughtful responses. They clarify my questions/concerns.

---

### Official Review · Reviewer_2vjz · 2026-03-13

**Soundness:** 3
**Presentation:** 3
**Significance:** 3
**Originality:** 3
**Overall Recommendation:** 4
**Confidence:** 2

**Summary:**

Multiple-choice question answering exposes models to answer options that can be exploited in ways that inflate reasoning ability. This work  study this phenomenon across 15 question-answering benchmarks and 27 LLMs by varying how and when models are exposed to answer options. Investigation reveals three lessons for the design and interpretation of LLM evaluations: (1) Decoupling is essential and increasingly infeasible for reasoning models.  (2) design for option-independent correctness: write stems that do not reference the options and either
define a canonical free-form answer or score via post-hoc mapping. (3) Relying solely on more challenging distractors as an antidote to exploitation is insufficient.

**Compliance With Llm Reviewing Policy:**

Affirmed.

**Final Justification:**

My concerns are addressed and I decide to keep my score.

**Key Questions For Authors:**

Is there a way to make the 40% of the original questions that were not "viable" for open-style answering become viable

**Limitations:**

It might be helpful to discuss the impact of removing nearly 40% of the original questions during the MCQA-to-FTQA conversion.

**Strengths And Weaknesses:**

Strengths
- The study evaluates a substantial number of 27 models across 15 different benchmarks, providing a broad and robust dataset for its conclusions.
- The research question is well-motivated
- The paper is well-organized

Weaknesses
- The process of converting MCQA to FTQA involved removing nearly 40% of the original questions because they were not "viable" for open-style answering. This could potentially bias the results toward certain types of reasoning tasks that are easier to convert
- The paper use some complex terminology eg. Q-CoT-MC-1T which might be difficulty for readers to keep track throughout the paper

---

> ### Author Rebuttal · Authors · 2026-03-27
>
> Thank you for the thoughtful review and for highlighting the conversion issue. Our goal in the conversion was to isolate the effect of format while avoiding any rewriting that might bias the question away from what was posed. For that reason, when converting MCQA benchmarks to FTQA, we removed questions whose stems explicitly or implicitly depended on the option set (e.g., "which of the following") as well as stems that were not well-formed standalone questions. However, we think its an interesting future direction to think about how one might design a conversion system at scale. For example, we think it is non-trivial to characterize what an accurate conversion to FTQA would be from the following MMLU MCQA question that preserves the original semantics:
>
>     Which of the following is true:
>     A. Every equivalence relation is a partial-ordering relation.
>     B. Number of relations form A = {x, y, z} to B= (1, 2), is 64.
>     C. Empty relation _ is reflexive
>     D. Properties of a relation being symmetric and being un-symmetric are negative of each other.
>
> We also appreciate the comment on terminology. We will add a visual schematic to the next draft to make the connections easier to follow.

---

> > ### Author Rebuttal · Reviewer_2vjz · 2026-04-03
> >
> > Thanks for the authors' response. I decide to keep my score.

---

### Decision · Program_Chairs · 2026-04-30

**Decision:**

Accept (regular)

**Comment:**

When evaluating Large Language Models (LLMs) in question-answering domains, multiple-choice question answering (MCQA) is widely used because it enables automatic grading. The authors argue that, for non-reasoning LLMs, MCQA remains a good proxy for free-text performance when any chain-of-thought is produced only before the options are revealed. However, this "decoupled" format is not realizable for most reasoning models: they are designed to emit reasoning tokens whenever they are prompted, so if options are present they inevitably "reason over" the options. In practice, this creates large, misleading gains over free-text baselines. On benchmarks where reasoning can be expressed as code, the authors ask models to output code, and execute it varying the inputs, and compare the resulting input–output behavior, revealing failure modes that MCQA diagnostics obscure. Lastly, we offer practical guidelines when analyzing results from MCQA that better reflect LLMs' genuine reasoning capabilities.

Reviewers appreciated the broad empirical coverage, evaluating 27 models across 15 benchmarks, spanning many modern LLM families and multiple scenarios involving MCQA with chain-of-thought reasoning. Reviewers also found the paper well written and easy to follow, with a strong introduction and clear organization.

Overall, the meta-reviewer recommends acceptance of the paper.